# Clustering in Causal Attention Masking

Nikita Karagodin[*]     Yury Polyanskiy[†]     Philippe Rigollet[‡]

## Abstract

This work presents a modification of the self-attention dynamics proposed by Geshkovski et al. (2023b) to better reflect the practically relevant, causally masked attention used in transformer architectures for generative AI. This modification translates into an interacting particle system that cannot be interpreted as a mean-field gradient flow. Despite this loss of structure, we significantly strengthen the results of Geshkovski et al. (2023b) in this context: While previous rigorous results focused on cases where all three matrices (Key, Query, and Value) were scaled identities, we prove asymptotic convergence to a single cluster for arbitrary key-query matrices and a value matrix equal to the identity. Additionally, we establish a connection to the classical Rényi parking problem from combinatorial geometry to make initial theoretical steps towards demonstrating the existence of meta-stable states.

## 1    Introduction

The introduction of the Transformer architecture Vaswani et al. (2017) has markedly impacted the landscape of natural language processing (NLP), signaling the advent of large language models. Central to the Transformer architecture is the *self-attention mechanism*, a special kind of layer that distinguishes it from preceding models such as ResNets. This innovation has yielded unprecedented performance not only in machine translation and text summarization but also in areas beyond NLP, including computer vision, speech recognition, and robotics. The flexibility and efficiency of Transformers underscore their integral role in the progression of artificial intelligence. Despite their widespread use, the theoretical foundations underlying their success remain underexplored.

Following Sander et al. (2022), recent studies by Geshkovski et al. (2023a) and Geshkovski et al. (2023b) have proposed a mathematical framework to analyze Transformers as interacting particle systems, demonstrating that tokens, when modeled as particles, exhibit clustering under certain conditions on the Key, Query, and Value matrices. These works primarily focus on full (mean-field) attention mechanisms, where each token can interact with every other token. Building upon this foundation, our research extends the analysis to *causal* attention mechanisms, wherein each token is restricted to interact only with preceding tokens. This distinction is crucial, as causal attention is prevalent in Transformer models employed in generative AI and known as decoder architectures.

Causal attention is crucial for sequence generation tasks, ensuring that each token only attends to previous tokens and not future ones, thereby preserving the correct temporal order. This mechanism, also known as autoregressive attention, masks future tokens during attention computation to prevent the model from accessing information it hasn't generated yet. At inference time, causal attention allows the model to generate text one token at a time, using previously generated tokens to inform the next, ensuring coherent and contextually accurate sequences. This step-by-step generation process is computationally efficient, as each token is produced in a forward pass without needing to revisit previous steps. In contrast to full attention, which considers all tokens simultaneously and is suitable

---

[*]Laboratory for Information and Decision Systems, MIT, Cambridge, MA, USA

[†]Laboratory for Information and Decision Systems, MIT, Cambridge, MA, USA

[‡]Department of Mathematics, MIT, Cambridge, MA, USA

38th Conference on Neural Information Processing Systems (NeurIPS 2024).

Table 1: Possible Final Configurations of Particles

| Largest Eigenvalue | Multiplicity | Final Configuration | Figure |
|:---:|:---:|:---:|:---:|
| $\lambda_{\max} > 0$ | d | First particle $x_0$ | 1a |
| $\lambda_{\max} > 0$ | $\geq 2$ | One point in $L$ | 1b |
| $\lambda_{\max} > 0$ | 1 | Two points $\xi$ and $-\xi$ | 1c |
| $\lambda_{\max} < 0$ | $\geq 2$ | Point cloud around $L$ | 1d |
| $\lambda_{\max} < 0$ | 1 | Two point clouds around $\xi$ and $-\xi$ | 1e |

for tasks like machine translation where the entire sequence is known, causal attention is essential for tasks requiring real-time, sequential output. This computational advantage explains the pervasiveness of causal attention not only in natural language processing but also in image generation with tools like DALL-E (Ramesh et al., 2021), VQGAN (Esser et al., 2021), or Parti (Yu et al., 2022) and multimodal foundation models, notably Chameleon (Team, 2024). More generally, the use of *masked* attention where tokens pay attention to a subset of other tokens has been driving recent scaling efforts and has led to state-of-the-art models such as MUSE (Chang et al., 2023) or Alphafold 3 (Abramson et al., 2024). Causal attention can also be recast as an interacting particle system but it requires different analytical techniques. This is the goal of our paper.

**Our contributions.** Our main theoretical result establishes asymptotic clustering of tokens for causal self-attention transformer modeled as an interacting particles system on the sphere (Theorem 4.1). While mathematically accurate, this asymptotic collapse to a single cluster is seldom observed numerically. Instead, particles collapse to multiple clusters and stay in this configuration for a very long time (see Fig. 2 for a representative example) — such *meta-stable* states were already alluded to in Geshkovski et al. (2023b) and their study was recently initiated in Koubbi et al. (2024). In Section 5 we describe such meta-stable states using analogy with the Rényi parking process (Lemma 2, Theorem 5.1). Additionally, Theorem 5.1 covers asymptotic clustering of tokens for causal self-attention with additional cross-attention component. Moreover, we predict that, akin to linear dynamical systems, the most important factors that qualitatively describe final particles configuration both in causal and full-attention cases are the eigenvalue of the Value matrix $V$ with the largest real part $\lambda_{\max}$ and its eigenspace $L$, while Query and Key matrices $Q, K$ and temperature parameter $\beta$ do not matter. Our conjectured atlas of possible meta-stable configurations is listed in Table 1. We prove the result stated as the first line of this table, namely that particles eventually collapse into a point when $V = I_d$ in Theorem 4.1. We remark that assumptions of Theorem 4.1 are much weaker than for the similar results in the full-attention case Geshkovski et al. (2023b), in particular we put no constraints on $K, Q$ or $\beta$. This work is a combination of rigorous mathematical results and non-trivial predictions based on analytical insights and numerical simulations. We summarize all limitations in Section 6.

**Related work.** Our work builds upon the framework of Geshkovski et al. (2023b,a) where clustering properties of transformers are analyzed as systems of particles interacting on the sphere. Specifically, Geshkovski et al. (2023b) proved that encoder-only (i.e. unmasked) self-attention with (post) LayerNorm leads to tokens clustering to a single point, in the limit of number of layers going to infinity. This phenomenon is also known as *consensus* in the related literature of multi-agent systems Markdahl et al. (2017); Criscitiello et al. (2024) and Kuramoto oscillators Strogatz (2000); Abdalla et al. (2022). Work Geshkovski et al. (2023b) in turn expands on the original perspective brought forward by Sander et al. (2022) that identify the self-attention layer as a measure-to-measure map, see also Vuckovic et al. (2021). More recently, Castin et al. (2024) studied the smoothness of this map in a framework that also covers causal attention. This work introduces a clever reparametrization that allows them to recast causal attention as mean-field dynamics, akin to their full attention counterpart. Using various approximations, Cowsik et al. (2024) were able to study a more realistic architecture that also includes MLP layers and produce accurate predictions for the final configuration of particles. This setup was further investigated by Agrachev and Letrouit (2024) from a geometric control perspective. We note also that clustering in the absence of a residual connection (replace $\dot{x}_k(t)$ with $x_k(t+1)$ in (SA)) was established in Wu et al. (2023). Additional effects of the residual connection are studied in Dong et al. (2021) and Zhang et al. (2024).

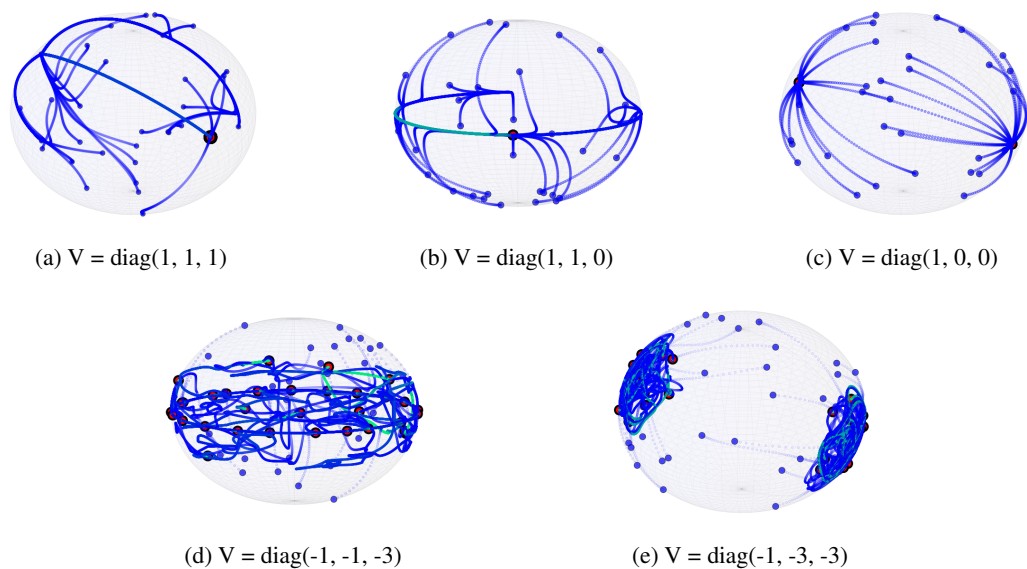

| (a) V = diag(1, 1, 1) | (b) V = diag(1, 1, 0) | (c) V = diag(1, 0, 0) |

| (d) V = diag(-1, -1, -3) | (e) V = diag(-1, -3, -3) |

Figure 1: Particle trajectories for different Value matrices. In all cases we take simple Query and Key matrices $K = Q = I_d$, temperature $\beta = 9$ and final time $T = 5000$ for $n = 32$ particles initialized uniformly at random on the sphere. Positions of particles at time $T$ are indicated by a red dot.

## 2 Causal attention

Before describing our model of causal attention dynamics, we review the idea of Geshkovski et al. (2023b) for modeling the *full attention* dynamics. In that work, the evolution of representations of tokens through the layers is modeled as a system of $n$ coupled Ordinary Differential Equations (ODEs) describing dynamics of a system of particles $x_1(t), \ldots, x_n(t)$. A brief part of their derivation of the dynamics from the transformers architecture is written in Section A.1. The particle position $x_k(t)$ corresponds to representation of the $k$-th *token* at layer $t$ (where for convenience, $t$ is allowed to take non-integer values) and due to *RMSNorm* the particles are forced to live on a unit sphere $\mathbb{S}^{d-1}$. (RMSNorm layer usually also includes a multiplication by a trainable diagonal matrix $D$, but the effect of this step can be equivalently achieved by multiplying $K, Q, V$ matrices by $D$.) These ODEs are parametrized by three matrices, known as the query $Q$, the key $K$ and the value $V$, respectively, and that are assumed to be square $d \times d$ matrices. More specifically, token $k$ evolves according to

$$\dot{x}_k(t) = \mathbf{P}_{x_k(t)} \Big( \frac{1}{Z_k(t)} \sum_{j=1}^n e^{\beta \langle Q x_k(t), K x_j(t) \rangle} V x_j(t) \Big), \tag{SA}$$

where $\mathbf{P}_x y = y - \frac{\langle x, y \rangle x}{|x|^2}$ is the projection onto the tangent space of $\mathbb{S}^{d-1}$ at $x$, and

$$Z_k(t) = \sum_{j=1}^n e^{\beta \langle Q x_k(t), K x_j(t) \rangle}$$

is a normalizing factor. Note that the dynamics of the $k$-th token depend on the positions of *all tokens* $j \in [n]$, which is a landmark characteristic of full attention leading to the so-called *mean-field dynamics* studied in Geshkovski et al. (2023b); see also Geshkovski et al. (2023a); Castin et al. (2024); Paul and Trélat (2024).

In this work we focus on *causal attention*, where the dynamics of token $k$ depend only on the position of tokens $j \leq k$. As described in the introduction, this modification is by now the dominant type of transformer architecture in generative AI. To reflect causal masking, we modify the ODE governing the dynamics of token $k$ as follows:

$$\dot{x}_k(t) = \mathbf{P}_{x_k(t)} \Big( \frac{1}{Z_k(t)} \sum_{j=1}^k e^{\beta \langle Q x_k(t), K x_j(t) \rangle} V x_j(t) \Big), \tag{CSA}$$

where the normalizing factor $Z_k(t)$ is naturally updated to

$$Z_k(t) = \sum_{j=1}^{k} e^{\beta\langle Qx_k(t), Kx_j(t)\rangle}.$$

## 3  Single token dynamics

Note that in (CSA) dynamics, the first token is evolving fully autonomously without the influence of others. Thus, we start from the description of its evolution. It will also guide our understanding of the dynamics of subsequent tokens. The first token moves according to the equation

$$\dot{x}(t) = \mathbf{P}_{x(t)}(Vx(t)).$$

To state its behavior for any matrix $V$, we need a few definitions. Denote $\lambda_{\max}$ as the largest real part of all the eigenvalues of $V$. Let $L'$ be the span of all generalized eigenvectors of $V$ associated to eigenvalues(potentially different) with their real part equal to $\lambda_{\max}$. Let $L \subset L'$ be the subspace generated by only the eigenvectors in $L'$ with the largest corresponding Jordan block (the vectors might correspond to different blocks and even to different eigenvalues).

**Lemma 1.** *Let $x(t)$ be a solution of an ODE $\dot{x}(t) = \mathbf{P}_{x(t)}(Vx(t))$ defined on the unit sphere $\mathbb{S}^{d-1}$. Then, for almost every initial value $x(0) \in \mathbb{S}^{d-1}$, there exists $C, c > 0$ such that the following convergence rates for the geodesic distance* **dist** *hold:*

*(i) Exponential convergence to $L'$:* **dist**$(x(t), L' \cap \mathbb{S}^{d-1}) \leq Ce^{-ct}$, *and*

*(ii) linear convergence to $L$:* **dist**$(x(t), L \cap \mathbb{S}^{d-1}) \leq ct^{-1}$

This result can be derived from standard results on the theory of linear ODEs (proof in Section B.1). We note that this result is important for other tokens as well. Indeed, for every token $x_k$, the contribution to $\dot{x}_k$ in (CSA) from the term with $j = k$ often has the biggest weight, an effect amplified by large $\beta$.

In general, eigenvectors corresponding to a real eigenvalue $\lambda = \lambda_{\max}$ create a fixed set in $L \cap \mathbb{S}^{d-1}$, while the complex eigenvalues with the largest real part produce a limit torus in $L \cap \mathbb{S}^{d-1}$. In what follows, we only consider the case where the eigenvalue with the largest real part is real itself and it only has Jordan blocks of size 1. Then, $L = L'$ and convergence to $L$ is exponentially fast. Note also that when $\dim L = 1$, we have $L \cap \mathbb{S}^1 = \{\pm\xi\}$ for some unit vector $\xi$. In this case, $x(t) \to \pm\xi$ as $t \to \infty$, again with exponential speed. These observations will be important for the next section, when we describe asymptotic configurations of tokens.

## 4  Final Configuration

The system of $n$ tokens that we are studying is far more complicated than for a single token. Even establishing convergence to *some* point as $t \to \infty$ is challenging. In Geshkovski et al. (2023b), similar models were analyzed analytically by noticing that the dynamical system has the structure of the gradient flow of some potential function:

$$\dot{x}(t) = \nabla H(x).$$

For such systems, groundbreaking results of Łojasiewicz (1962, 1965, 1984) (see Haraux (2012) for a self-contained overview) guarantee convergence to a critical point of $H$ assuming it is real-analytic.

However, our system (CSA) does not have a gradient-flow structure and thus techniques of Łojasiewicz are not applicable. On the other hand, we have a significant advantage in the hierarchical structure of our system, allowing us to study tokens sequentially.

We have already understood the evolution of the first token. In this section, we do two things. First, we describe, based on our analytical and numerical insights, conjectures about the asymptotic configuration $x(t)$ for $t \to \infty$. The surprising result here is that only the spectral properties of $V$ (and not $K$ or $Q$) affect asymptotics. Second, we rigorously prove convergence to a single point for the special case of $V = I_d$. We note that unlike the proof in Geshkovski et al. (2023b) (see also Markdahl

et al. (2017); Criscitiello et al. (2024)), our result works for all $K$ and $Q$ matrices, while the proof in Geshkovski et al. (2023b) works only for $Q^\top K = V$ and Markdahl et al. (2017); Criscitiello et al. (2024) is restricted to $Q^\top K = I_d$.

Our main insight is that there are two major forces that drive each token: its internal force which is described by Lemma 1, and the external force induced by all the particles preceding it, which is either attractive or repulsive depending on the sign of the top eigenvalue(s) of $V$. The balance between the two forces is defined via attention.

To get a better grasp of how the external force works, we consider the case where the first (internal) force vanishes, that is, $V = I_d$. In this case, the tokens collapse asymptotically to a single point.

**Theorem 4.1.** *Let $V = I_d$ and $Q, K$ be arbitrary matrices. Then, for almost any starting point $(x_1(0), \ldots, x_n(0))$ with respect to the volume measure on $(\mathbb{S}^{d-1})^n$, the causal transformer dynamics (CSA) converge to a single cluster:*

$$\forall\, k \in [n], \ \lim_{t \to \infty} x_k(t) = x_1(0).$$

We prove this result in Section B.2. In the proof, weight functions are only required to be positive and continuously differentiable ($C^1$). This ambiguity suggests that incorporating time-dependence of $Q$ and $K$ might not alter the theorem's validity, but it significantly adds complexity to the proof in dealing with non-autonomous systems.

Steps similar to our proof of Theorem 4.1 can be followed to study the more general case of the matrix $V \neq I_d$. Unfortunately, one runs into multiple technical issues with application of the stable-manifold theorem from dynamical systems due to the emergence of critical manifolds (as opposed to critical points in the $V = I_d$ case). Thus, we leave the general case at the status of conjectures, which we describe next.

In what follows, we denote the eigenvalue of $V$ with the largest real part as $\lambda_{\max}$ and assume that it is real. If it is not, the limiting configuration is additionally rotating with a constant speed, which complicates the discussion and so is omitted.

Let $L$ denote the eigenspace of $\lambda_{\max}$. If $\lambda_{\max}$ has multiplicity 1, then we denote the corresponding unit eigenvectors as $\pm \xi$. For simplicity we assume that $\lambda_{\max}$ has all of the corresponding Jordan blocks of size 1.

First of all, if $\dim L = 1$ then, according to Lemma 1, every token is driven towards $\xi$ or $-\xi$ by their own force. Moreover, for $\lambda_{\max} > 0$ the force of other tokens is attractive, while for $\lambda_{\max} < 0$ it is repulsive.

Thus, for $\lambda_{\max} > 0$ all the particles collapse into $\xi$ and $-\xi$, whereas for $\lambda_{\max} < 0$ the repulsion force prevents the particles from going all the way to $\pm \xi$ and instead the particles stabilize at two clouds around $\xi$ and $-\xi$. This behavior is captured in Figures 1c and 1e[4]. For the case $\lambda_{\max} > 0$ we formally express it as:

**Conjecture 1.** *Let $Q, K$ be arbitrary matrices and $V$ be diagonalizable with d different positive real eigenvalues. Denote the largest eigenvalue as $\lambda_{\max}$ and unit vector $\xi : V\xi = \lambda_{\max}\xi$. Then, for almost any starting point $(x_1(0), \ldots, x_n(0))$ with respect to the volume measure on $(\mathbb{S}^{d-1})^n$, the causal transformer dynamics (CSA) converge to two clusters*

$$\forall\, k \in [n], \ \lim_{t \to \infty} x_k(t) \in \{\xi, -\xi\}.$$

If $\lambda_{\max}$ has multiplicity at least 2, then from Lemma 1 each token internally gets attracted by the eigenspace $L$. When tokens are close to $L$, the action of $V$ becomes close to $\lambda_{\max}I_d$, which for $\lambda_{\max} > 0$ according to Theorem 4.1 forces tokens to collapse to a singleton, while for $\lambda_{\max} < 0$ other tokens exude a repelling force, causing particles to spread out around $L$. This behavior is captured in Figures 1b and 1d. For the case $\lambda_{\max} > 0$ we formalize it as follows:

**Conjecture 2.** *Let $Q, K$ be arbitrary matrices and $V$ be any matrix such that its largest eigenvalue $\lambda_{\max} > 0$ is real and has an eigenspace $L$ of dimension $\dim L \geq 2$, while for any $z \in L^\perp$ one*

---

[4]The spread of the clouds depends on the relative importance of each token's own attention, that differs with various $K, Q$. There are choices of $K$ and $Q$ that result in complex interactions without structure. For example, when $Q^\top K = [[0, -1, 0], [1, 0, 0], [0, 0, 1]], V = -I_3$

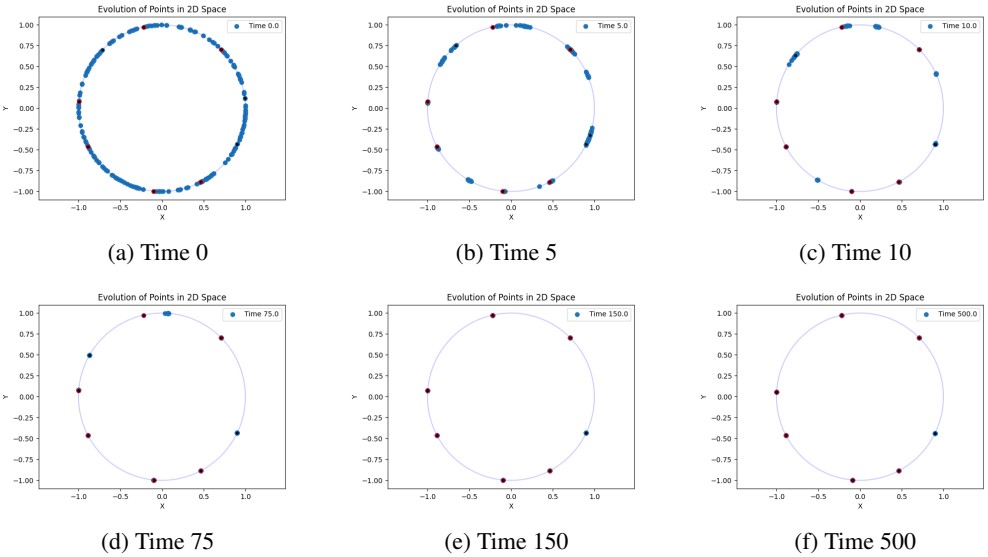

| (a) Time 0 | (b) Time 5 | (c) Time 10 |
|---|---|---|
| (d) Time 75 | (e) Time 150 | (f) Time 500 |

Figure 2: Evolution of the system (CSA) with $K = Q = V = I_2$ with $n = 200$, $d = 2$, $\beta = 64$, strong Rényi centers (red) and Rényi centers (black) with $\delta = 4\beta^{-1/2}$. Note that strong Rényi centers are visually stationary (as per Lemma 2) but do not explain all clusters. In turn, Rényi centers are moving and merging (one disappears between $t = 75$ and $t = 150$), but capture more meta-stable clusters.

has $Vz \in L^\perp$ with $\langle Vz, z \rangle < \lambda_{\max}|z|^2$. Then, for almost any initialization $(x_1(0), \ldots, x_n(0))$, the causal attention dynamics (CSA) converge to one cluster. More specifically, if we define $\xi$ as the normalized $L$-component of $x_1(0)$, i.e., for $y_1 := \mathbf{P}_{L^\perp}(x_1(0))$, $\xi := y_1/|y_1|$, then

$$\forall\, k \in [n],\ \lim_{t \to \infty} x_k(t) = \xi.$$

(Note that $\xi$ is undefined when $x_1(0) \perp L$, but this happens with probability zero.)

An important practical observation is that these conjectures explain that $V$ performs dimensionality reduction in the following way. Tokens converge to $L \cap \mathbb{S}^{d-1}$ and, in that space, they move as if acted upon by the $\lambda I_d$ matrix on a sphere $\mathbb{S}^{\dim L-1}$. For the pre-trained Lan et al. (2020) the spectra of value matrices is depicted in Figure 4. Interestingly, there are heads with negative $\lambda_{\max}$. Future work will be concerned with studying real-world matrices $V$ and connecting their top eigenspaces to semantic meaning of layers and tokens.

## 5 Meta-stable clustering

As we discussed earlier, perhaps the most fascinating discovery of Geshkovski et al. (2023b) is the existence of meta-stable clusters in the full-attention dynamics. It turns out that the same phenomenon persists in the causal-attention dynamics that we study here.

The dynamical evolution of the system is illustrated in Fig. 2. At $t = 150$, the initially uniform distribution of 200 particles consolidates into seven distinct clusters. While Theorem 4.1 establishes the eventual collapse into a single cluster, these intermediate clusters exhibit remarkable metastability, persisting with negligible movement over extended time periods—at least until $t = 500$ according to Fig. 2—before sequential merging occurs. We define these meta-stable configurations as meta-stable clusters, with three-dimensional analogues shown in Fig. 1a and 1b.

Given that the time parameter in our dynamics corresponds to network depth in transformer architectures, the meta-stable configurations, rather than final states (achieved at $t = \exp(\Omega(\sqrt{\beta}))$), hold greater practical significance. The emergence of meta-stable clustering and its associated dimensionality reduction may provide fundamental insights into transformers' capacity for generating efficient context-dependent representations.

From a theoretical perspective, understanding meta-stable clustering presents significant challenges, as traditional techniques for asymptotic analysis—such as those used in our Theorem 4.1—prove insufficient. Recent work on full attention transformers has made partial progress in this direction. Koubbi et al. (2024) demonstrated that when self-attention dynamics approach a nearly clustered state, they will converge to a tightly clustered configuration and remain stable for an exponential time period. Complementing this, Bruno et al. (2024) proved that tokens initialized near a uniform distribution on the sphere will spontaneously organize into a loosely clustered state. However, the bounds on the clustering tightness in this second line of work are not sufficient to trigger the convergence conditions required by Koubbi et al. (2024)'s theorem.

This Section presents a fundamental discovery regarding the identification of cluster centers in causal-attention dynamics. We establish three key claims: First, we demonstrate that initialization irregularities generate distinctive tokens, termed Rényi parking centers, which evolve into meta-stable cluster nuclei. While this phenomenon is primarily supported by numerical evidence (Fig. 3), it provides crucial insight into the clustering mechanism. Second, we prove that a subset of these special tokens, called strong Rényi centers, maintains near-stationarity over extended time periods (Lemma 2). Both Rényi and strong Rényi centers occur with frequency $\Theta(\beta^{\frac{d-1}{2}})$, confirming the $\sqrt{\beta}$ scaling predicted for $d = 2$ by Geshkovski et al. (2023b); see also Koubbi et al. (2024); Bruno et al. (2024). Third, we establish in Theorem 5.1 that as $t \to \infty$, all remaining tokens will converge to the vicinity of one of these stationary tokens, completing the meta-stable clustering process.

This section restricts our analysis to the case where $V = I_d$. For general matrices $V$, our empirical observations suggest that particles rapidly converge to a lower-dimensional subspace spanned by $d_1 \ll d$ principal eigenvectors. Consequently, we conjecture that the number of meta-stable clusters should rather be $\beta^{\frac{d_1-1}{2}}$, where the ambient dimension $d$ is replaced by the effective dimension $d_1$. While a rigorous proof of this dimension-reduction remains an open problem for future investigation, this phenomenon motivates our focus on low-dimensional cases (specifically $d = 2$) throughout this section.

For convenience, we also fix $Q = K = I_d$, though this condition could be easily relaxed (e.g. to $Q^\top K = K^\top Q = V$). Under these assumptions, the system can be rewritten in polar coordinates $x_k = [\cos(\varphi_k), \sin(\varphi_k)]^\top$ as

$$\dot{\varphi}_k = \frac{1}{Z_k} \sum_{j=1}^{k} e^{\beta(\cos(\varphi_k - \varphi_j)-1)} \sin(\varphi_j - \varphi_k) = \frac{1}{Z_k} \sum_{j=1}^{k-1} h(\varphi_j - \varphi_k), \qquad \text{(CSA-2d)}$$

with interaction potential given by

$$h(x) := e^{\beta(\cos(x)-1)} \sin x, \quad \text{and} \quad Z_k = \sum_{j=1}^{k} e^{\beta(\cos(\varphi_k - \varphi_j)-1)}. \qquad \text{(IntPot)}$$

## 5.1 Rényi Parking

The prediction of meta-stable clustering center locations exhibits a notable connection to the Rényi parking problem.

Consider a sequence of tokens $(x_j)_{j \geq 1}$ on the sphere $\mathbb{S}^{d-1}$ equipped with geodesic distance $\mathsf{dist}$. For a fixed separation parameter $\delta > 0$, we define:

- *Rényi centers* as the subsequence $(x_{s_j})_{j \geq 1}$, where $(s_j)_{j \geq 1}$ is strictly increasing and satisfies:

$$\mathsf{dist}(x_{s_j}, x_{s_i}) > \delta \quad \text{for all } i < j$$

- *Strong Rényi centers* as the subsequence $(x_{s_j})_{j \geq 1}$ satisfying:

$$\mathsf{dist}(x_{s_j}, x_i) > \delta \quad \text{for all } i < s_j$$

By construction, the set of Strong Rényi centers forms a subset of Rényi centers.

As demonstrated in Section A.2, particles in our system exert maximal attractive force at distances of order at most $\beta^{-1/2}$, with rapid decay beyond this scale. For strong Rényi centers defined

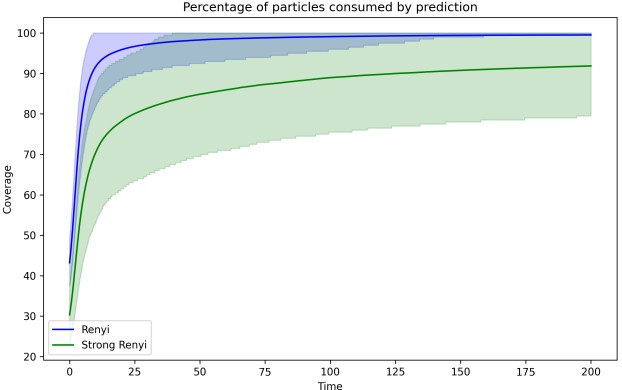

Figure 3: Total percentage of particles consumed by Rényi and strong Rényi centers over time. Here we have plotted average, $0.1$ and $0.9$ quantiles over 5000 experiments with $n = 200$, $d = 2$, $\beta = 64$, $\delta = 4\beta^{-1/2}$.

with separation parameter $\delta = c\beta^{-1/2}$ (where $c$ is sufficiently large), this decay ensures negligible influence from preceding particles and thus remain stable for a long time—a phenomenon formally established in Lemma 2. This metastability, coupled with rapid particle aggregation tokens, indicates that strong Rényi centers serve as primary attractors for subsequent tokens.

Rényi centers are unaffected by previous particles but only by previous Rényi centers, thereby generating new clustering centers. For fixed $\delta$, there are more Rényi centers than strong Rényi centers (see Section C.4 for exact cardinality analysis). While Rényi centers better capture the meta-stable clustering effect, as illustrated in Figures 3 and 2, they lack positional stability and may converge to other centers over time. Although Rényi centers rapidly aggregate a large fraction of particles, some of these particles continue to migrate and eventually converge to strong Rényi centers.

The next result shows that strong Rényi centers remain nearly fixed for a long time.

**Lemma 2.** *Let $d = 2$ and $Q = K = V = I_2$. Consider a subsequence of strong Rényi centers $x_{s_1}, \ldots, x_{s_m}$ satisfying the separation condition with constants $\varepsilon, c > 0$*

$$\min_{i < s_j} |x_{s_j} - x_i| > c(1 + 2\varepsilon)\beta^{-1/2}.$$

*Assume that*

$$c > \beta^{1/2} \arccos((-1 + \sqrt{4\beta^2 + 1})/(2\beta)). \tag{1}$$

*Then for any time $T_j$ such that*

$$T_j s_j h(c\beta^{-1/2}) < \varepsilon c\beta^{-1/2}, \tag{2}$$

*where the interaction potential $h$ is defined in* (IntPot)*, the displacement of each center is bounded by*

$$\max_{t \in [0, T_j]} |x_{s_j}(t) - x_{s_j}(0)| < \varepsilon c\beta^{-1/2}.$$

The key observation driving this result is that strong Rényi centers are weakly affected by all previous particles. However, though this is correct on short time scales, it should be checked for all times in $[0, T]$. A complete proof can be found in Section C.1.

**Remark 1.** *Using the properties of $h$ derived in Section A.2 it can be shown that*

- *For $\beta > 1$ a sufficient condition for* (1) *to hold is simply $c > 1$,*

- *a sufficient condition for* (2) *to hold is*

$$T_j s_j < e^{c^2/2 - c^4/(24\beta)}\varepsilon.$$

*Moreover, it is easy to prove that indexes $s_j$ are mostly small. Thus, we see that early strong Rényi centers are almost stationary for a time that is exponential with the square of separation magnitude.*

Rényi centers and strong Rényi centers play a fundamental role in meta-stable clustering, warranting analysis of their properties as extreme points in a sequence. While defined here using geodesic distance on a sphere, the definition extends naturally to distances induced by $\langle Qx, Ky \rangle$ under appropriate conditions. This generalization aligns with our observation that meta-stable clustering occurs in the subspace $L$ where $V$ sends tokens and acts as the identity on $L$.

The distribution of these centers under various initialization schemes presents a key analytical challenge. Section C.4 addresses the uniform i.i.d. case, where Rényi's classical result characterizes the expected number of centers. Extensions to general distributions and Markov processes—more relevant to language processing applications—remain open for Rényi centers due to their structural complexity, particularly in higher dimensions ($d > 2$). In contrast, strong Rényi centers are much easier to handle: even our computation of the average number of centers in Section C.4 works for any distribution regardless of the dimension.

## 5.2 Fixed Meta-stable Clustering Centers

Having established the existence of $O(\sqrt{\beta})$ quasi-stationary tokens for $d = 2$ and $n \gg 1$, we next examine their role as cluster centers. While Figures 3 and 2 provide substantial numerical evidence that these tokens attract and aggregate nearby particles, a rigorous proof remains elusive. We establish instead a weaker result: when quasi-stationary tokens are artificially frozen (analogous to cross-attention in encoder-decoder architectures), all other tokens converge to these frozen centers. This simplified model, while instructive, differs from true meta-stable clustering in important aspects detailed in Section 6.

We only state our result for $d = 2$ and identity parameter matrices as in (CSA-2d).

**Theorem 5.1** (Clustering to frozen tokens for $K = Q = V = I_2$). *Let $\theta_1, \ldots, \theta_m$ be fixed tokens that are well-separated, namely $|\theta_i - \theta_j| > c\beta^{-1/2}$. Let $\mu_0$ be an absolutely continuous probability measure on $(\mathbb{S}^1)^n$ and let $\varphi_1(0), \ldots, \varphi_n(0) \sim \mu_0$. Consider causal attention dynamics (CSA-2d), with additional influence from the fixed tokens $\theta_j$, which enter evolution with additional weights $a_j \geq 1$. Specifically, we have*

$$\dot{\varphi}_k = \frac{1}{Z_k} \Big( \sum_{j=1}^{k} e^{\beta(\cos(\varphi_k - \varphi_j) - 1)} \sin(\varphi_j - \varphi_k) + \sum_{j=1}^{m} a_j e^{\beta(\cos(\varphi_k - \theta_j) - 1)} \sin(\theta_j - \varphi_k) \Big),$$

*with*

$$Z_k = \sum_{j=1}^{k} e^{\beta(\cos(\varphi_k - \varphi_j) - 1)} + \sum_{j=1}^{m} a_j e^{\beta(\cos(\varphi_k - \theta_j) - 1)}.$$

*Define $N = n + \sum_{j=1}^{m} a_j$ and $g = h'$ where $h$ is the interaction potential of* (IntPot).

*If $N, \beta, \varepsilon > 0$, and $c > 2 + 2\varepsilon$ satisfy:*

$$Nh((c - 1 - 2\varepsilon)\beta^{-1/2}) < h(\varepsilon\beta^{-1/2})$$
$$Ng((c - 2\varepsilon)\beta^{-1/2}) < g(\varepsilon\beta^{-1/2}),$$

*then with probability one, $\varphi(t)$ converges to an asymptotically stable critical point $\varphi^* \in (\mathbb{S}^1)^n$ satisfying:*

$$\forall k \in [n], \exists j \in [m] : |\varphi_k^* - \theta_j| < \varepsilon\beta^{-1/2}.$$

Since our dynamical system is not a gradient flow, the classical Łojasiewicz convergence theorem does not apply. Instead, we establish convergence by observing that the causal dynamics (both with and without frozen tokens) is, in fact, a *sequential* gradient flow, where each particle minimizes a slightly different energy potential. For such systems on $\mathbb{S}^1$, we demonstrate convergence through an alternative approach that circumvents the Łojasiewicz framework.

**Lemma 3.** *Consider a system of $n$ particles on $\mathbb{S}^1$ with angular coordinates $\varphi_1, \ldots, \varphi_n \in 2\pi\mathbb{R}/\mathbb{Z}$ evolving according to*

$$\dot{\varphi}_k = -\frac{1}{Z_k(\varphi_1, \ldots, \varphi_k)} \frac{\partial}{\partial \varphi_k} E_k(\varphi_1, \ldots, \varphi_k),$$

*where $E_1, \ldots, E_n$ are $C^1$ energy functions and $Z_k$ are $C^1$ normalization factors bounded by $0 < c < Z_k(\varphi) < C$. Assume:*

1. *Each $E_k$ has isolated critical points in $\varphi_k$ for fixed $\varphi_j, j \neq k$ (satisfied by analyticity),*

2. *For any $k \in [n]$, critical points restricted to the first $k$ particles are either strongly stable (the Jacobian has only eigenvalues with strictly negative real parts) or strongly unstable (there is an eigenvalue with a strictly positive real part).*

*Then for almost every initial condition $\varphi(0)$ with respect to Lebesgue measure, $\varphi(t)$ converges to a strongly stable critical point $\varphi^*$.*

The proof is deferred to Section C.2.

**Remark 2.** *The conditions of Theorem 5.1 are satisfied under the following explicit bounds:*

$$\varepsilon < 0.1, \quad c \geq 5.5 + 2\varepsilon, \quad \beta \geq (c - 1 - 2\varepsilon)^2/2, \quad N \leq \frac{\varepsilon}{c-1} \exp(3(c - 1 - 2\varepsilon)^2/8).$$

*Note that this result requires only $\beta \gtrsim \log N$. For example, taking $\varepsilon = 0.1$ and $c = 6.5$ yields $\beta \geq 14$ and $N \leq 700$ is sufficient. See Lemma 5 for the proof.*

## 6   Limitations

Our analysis presents both theoretical and practical limitations. From a theoretical perspective, we establish two key results: (1) strong Rényi centers maintain quasi-stationarity for time scales of order $\exp(c^2/2)$ per Lemma 2, and (2) exactly stationary centers attract all remaining particles (Theorem 5.1). However, this falls short of proving meta-stable clustering, as Theorem 5.1 provides no bounds on the convergence time. Consequently, we cannot guarantee that strong Rényi centers remain sufficiently stationary during particle aggregation. A complete meta-stability theory would require demonstrating that each Rényi center captures $\Omega(n)$ particles in $O(1)$ time as $n \to \infty$. Currently, even the weaker claim of capturing $\omega(1)$ particles remains unproven, presenting a crucial direction for future research.

The practical limitations stem from two model simplifications: the use of tied weights across layers (though supported by successful implementations, see Lan et al. (2020)), and the omission of the MLP layer central to Transformer architectures. Incorporating the MLP dynamics into our theoretical framework remains a significant open challenge.

## Acknowledgments

The work of NK and YP was partially supported by the MIT-IBM Watson AI Lab and the National Science Foundation under Grant No CCF-2131115. PR is supported by NSF grants DMS-2022448 and CCF-2106377, and a gift from Apple.

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

# A  Supplementary Material

## A.1  From Transformers to ODEs

The derivation of the equation (SA) was thoroughly described in Geshkovski et al. (2023b), but for completeness, we briefly repeat it here to explain how the problem arises.

In general, a typical Transformer architecture consists of repeated layers of multi-head attention, multi-layer perceptrons (MLP), normalization, and residual connections Vaswani et al. (2017). In this work, we simplify this setting by focusing only on the geometric behavior of a single-head attention layer with normalization and residual connections, omitting the MLP for brevity.

One head of a standard attention layer is defined as follows. Given an input sequence represented by the token embeddings $X \in \mathbb{R}^{n \times d}$, where $n$ is the number of tokens and $d$ is the dimension of the embedding space, and matrices $W_Q, W_K, W_V$ to compute queries, keys, and values, the attention mechanism computes a weighted sum of values based on their relevance to a query in the following form

$$\text{Attention}(X) = \text{softmax}\left(\frac{XW_Q W_K^\top X^\top}{\sqrt{d}}\right) XW_V.$$

By adding an RMS normalization from Zhang and Sennrich (2019) and a residual connection, the transformation from layer $t$ to layer $t + 1$ is given by:

$$X^{t+1} = X^t + \text{RMSNorm}(\text{Attention}(X^t)). \tag{3}$$

Here, different tokens are represented as rows of the matrix $X$ for computational reasons. For consistency with the convention that vectors are represented as columns, we transpose everything and denote a sequence of tokens encoded as particles in the $d$-dimensional embedding space $\mathbb{R}^d$ as $(x_1, \ldots, x_n)$, corresponding to the columns of $X^\top$. Additionally, to simplify the notation we denote $V := W_V^\top$, $Q := W_Q^\top$, and $K := W_K^\top$, and introduce an arbitrary temperature parameter $\beta$ instead of the fixed scaling factor $1/\sqrt{d}$. With these notational adjustments, one term of attention added to the $k$-th token can be written explicitly as:

$$\text{attn}(x_1, \ldots, x_n)_k = \frac{1}{Z_k} \sum_{j=1}^n e^{\beta \langle Qx_k, Kx_j \rangle} V x_j,$$

where

$$Z_k = \sum_{j=1}^n e^{\beta \langle Qx_k, Kx_j \rangle}.$$

The equation (3) can be interpreted as a discrete derivative, with $X^{t+1} - X^t$ representing the difference between layers. Therefore, the trajectory $X^t$ can be viewed as a discretization of a continuous flow. RMS normalization ensures that tokens remain on the scaled unit sphere, but from properly rescaling $Q, K, V$ we can assume that they stay on the standard unit sphere $\mathbb{S}^{d-1}$. Combining all these observations, the dynamics of token propagation through layers can be expressed as:

$$\dot{x}_k(t) = \frac{1}{Z_k(t)} P_{x_k(t)} \left( \sum_{j=1}^n e^{\beta \langle Q(t)x_k(t), K(t)x_j(t) \rangle} V(t) x_j(t) \right),$$

with

$$Z_k(t) = \sum_{j=1}^n e^{\beta \langle Q(t)x_k(t), K(t)x_j(t) \rangle},$$

and the projector $P_x(y) := y - \langle x, y \rangle x/|x|^2$ ensuring that $x_k$ remains on the sphere. This leads to the equation (SA), and applying a causal constraint, where each token attends only to the previous ones, transforms it into the causal attention equation (CSA) studied in this work.

## A.2  Interaction Potential

For completeness, here we describe the key properties of the interaction potential $h(x) = e^{\beta(\cos(x)-1)} \sin(x)$ from (IntPot), which defines the interactions between particles, and its derivative $g(x) = h'(x)$.

**Lemma 4** (Properties of Interaction Functions)**.** *Let $h(x) = e^{\beta(\cos x - 1)} \sin x$ and $g(x) = h'(x)$.*
*Then:*

1. *$h(x)$ is odd and $g(x)$ is even.*

2. *$h(x)$ is increasing on $[0, \tau_\beta^*]$ and decreasing on $[\tau_\beta^*, \infty)$, where*

$$\cos \tau_\beta^* = \frac{-1 + \sqrt{4\beta^2 + 1}}{2\beta}$$

   *and for $\beta \geq 1$,*

$$(\beta + 1/2)^{-1/2} < \tau_\beta^* < \beta^{-1/2}$$

3. *For $x > 0$, $h(x)$ is bounded by*

$$e^{-\beta x^2/2}(x - x^3/6) < h(x) < e^{-\beta x^2/2 + \beta x^4/24} x$$

4. *For $g(x)$, the following bounds hold:*

$$g(x) > e^{-\beta x^2/2}(1 - x^2/2 - \beta x^2) \qquad \text{for } 0 < x < (\beta + 1/2)^{-1/2}$$
$$g(x) > -e^{-\beta x^2/2 + \beta x^4/24} \beta x^2 \qquad \text{for } x > 0$$

*Proof.* 1. The oddness of $h$ and evenness of $g$ follow directly from their definitions.

2. Computing $g(x)$ explicitly:

$$g(x) = e^{\beta(\cos(x)-1)}(-\beta \sin^2(x) + \cos(x)) = e^{\beta(\cos(x)-1)}(\beta \cos(x)^2 + \cos(x) - \beta)$$

The sign of $g(x)$ changes at $\tau_\beta^*$, where

$$\cos \tau_\beta^* = \frac{-1 + \sqrt{4\beta^2 + 1}}{2\beta}$$

establishing that $h(x)$ increases on $[0, \tau_\beta^*]$ and decreases on $[\tau_\beta^*, \infty)$.

For the the lower bound on $\tau_\beta^*$: For $x < (\beta + 1/2)^{-1/2}$, we have

$$\cos x > 1 - x^2/2 > \beta x^2 > \beta \sin^2 x$$

implying $g(x) > 0$ in this region.

For the upper bound on $\tau_\beta^*$, fix $z = \beta^{-1/2}$ and observe that it suffices to show

$$\cos \tau_{\beta^*} = \frac{-1 + \sqrt{4\beta^2 + 1}}{2\beta} > \cos \beta^{-1/2} = \cos z$$

Using $\cos z < 1 - z^2/2 + z^4/24$, this reduces to verifying

$$-z^2/2 + \sqrt{1 + z^4/4} > 1 - z^2/2 + z^4/24$$

which holds for $z < 3.13$, satisfied when $\beta = z^{-2} \geq 1$.

3. and 4. The bounds on $h$ and $g$ follow from the standard inequalities

$$x - x^3/6 < \sin x < x, \quad \text{and} \quad -x^2/2 < \cos x < -x^2/2 + x^4/24$$

combined with our characterization of $g$'s sign via $\tau_\beta^*$. $\qquad \square$

We now turn to the proof of Remark 2.

**Lemma 5.** *Let $N$, $c$, $\varepsilon$, and $\beta$ satisfy:*

$$c \geq 5.5 + 2\varepsilon$$
$$\beta \geq (c - 1 - 2\varepsilon)^2/2$$
$$N < e^{3(c-1-2\varepsilon)^2/8} \frac{\varepsilon}{c - 1}$$

*Then:*

$$Nh((c - 1 - 2\varepsilon)\beta^{-1/2}) < h(\varepsilon\beta^{-1/2}) \quad \text{and} \quad -Ng((c - 2\varepsilon)\beta^{-1/2}) < g(\varepsilon\beta^{-1/2})$$

*Proof.* Let $r := c - 1 - 2\varepsilon$. From the assumptions, we have $r \geq 4.5$, $\beta \geq r^2/2 > 10$, and $\varepsilon < 0.1$. We must verify:

$$N < \frac{h(\varepsilon\beta^{-1/2})}{h(r\beta^{-1/2})}$$

$$N < \frac{g(\varepsilon\beta^{-1/2})}{-g((r+1)\beta^{-1/2})}$$

Using the bounds from Lemma 4, these inequalities reduce to:

$$N < \frac{\exp(-\varepsilon^2/2)\varepsilon\beta^{-1/2}(1 - \varepsilon^2/(6\beta))}{\exp(-r^2/2 + r^4/(24\beta))r\beta^{-1/2}}$$

and

$$N < \frac{\exp(-\varepsilon^2/2)(1 - \varepsilon^2/(2\beta) - \varepsilon^2)}{\exp(-(r+1)^2/2 + (r+1)^4/(24\beta))(r+1)^2}$$

Given $N < \exp(3r^2/8)\varepsilon/r$, it suffices to verify:

1. First inequality: Taking logarithms and using $\varepsilon < 0.1$, $\beta > 10$, we need:

$$-\frac{r^2}{8} + \frac{r^4}{24\beta} + \frac{\varepsilon^2}{2} < -\frac{3}{200}$$

Since $\beta \geq r^2/2$, this follows from:

$$-\frac{r^2}{8} + \frac{r^2}{12} < -\frac{1}{50}$$

which holds for $r \geq 4.5$.

2. Second inequality: After simplification using $\beta \geq r^2/2$, $\beta \geq 10$, $\varepsilon < 0.1$, we need:

$$f(r) = \frac{3r^2}{8} - \frac{(r+1)^2}{2} + \frac{(r+1)^4}{12r^2} + \frac{1}{200} + 2\ln(r+1) - \ln(9.8r) < 0$$

The derivative

$$f'(r) = \frac{1}{3}r - 1 + \frac{6r^3}{(r+1)^3(r-1)} + \frac{2}{r+1} - \frac{1}{r} > 0$$

for $r > 4.5$, as $r > 3$ and $2r > r + 1$. Therefore, it suffices to verify $f(4.5) \approx -4.14 < 0$. $\qquad\square$

## B   Final configuration

### B.1   Proof of Lemma 1

Let us show that trajectories $x(t)$ of our system can be characterized as normalized solutions of a linear homogeneous ODE in $\mathbb{R}^d$. Consider a solution $y(t)$ of:

$$\dot{y}(t) = Vy(t), \qquad y(0) = x(0) \tag{4}$$

For $s(t) := y(t)/\|y(t)\|$, we derive:

$$\dot{s}(t) = \frac{\dot{y}(t)}{\|y(t)\|} - \frac{y(t)}{\|y(t)\|^2}\left\langle \frac{y(t)}{\|y(t)\|}, \dot{y}(t) \right\rangle = Vs(t) - \langle s(t), Vs(t)\rangle s(t) = \mathbf{P}_{s(t)}(Vs(t))$$

Thus $x(t) \equiv s(t) = y(t)/\|y(t)\|$.

The solution to (4) has the following explicit form. Let $\{J_k\}$ denote the Jordan blocks of $V$ with: sizes $n_k$, eigenvalues $\lambda_k$, and generalized eigenvectors $\{\xi_1^k, \ldots, \xi_{n_k}^k\}$.

Then:

$$y(t) = \sum_k e^{\lambda_k t} \sum_{j=1}^{n_k} c_j^k \sum_{i=1}^{j} \frac{t^{j-i}}{(j-i)!}\xi_i^k \tag{5}$$

where the coefficients $\{c_j^k\}$ satisfy:

$$\sum_k \sum_{j=1}^{n_k} c_j^k \xi_j^k = x(0)$$

For almost all initial conditions $x(0)$ with respect to the surface measure on the sphere, all coefficients $c_j^k$ are non-zero. For complex eigenvalues $\lambda_k$, we combine conjugate terms to obtain real-valued solutions involving trigonometric functions.

The asymptotic behavior follows from two observations: (i) Terms with largest $\Re(\lambda_k)$ dominate as $t \to \infty$, corresponding to convergence to $L' \cap S^{d-1}$ at exponential rate, and (ii) Among these terms, those with highest power of $t$ (i.e., $t^{n_k-1} \xi_1^k$ terms) determine the slower convergence to $L \cap S^{d-1}$

### B.2 Proof of Theorem 4.1

We begin with a simple geometric lemma.

**Lemma 6.** *Let $x, y, z \in \mathbb{R}^d$ with $\|x\| = \|y\| = 1$. If $|\langle y, z \rangle| \le \langle x, z \rangle$, then:*

$$\langle x, z \rangle \ge \langle x, y \rangle \langle y, z \rangle$$

*with equality if and only if either: (i) $\langle x, z \rangle = 0$, or (ii) $|\langle y, z \rangle| = \langle x, z \rangle$ and $\langle x, y \rangle = sign(\langle y, z \rangle)$.*

*Proof.* By the Cauchy-Schwarz inequality and the hypothesis:

$$\langle x, y \rangle \langle y, z \rangle \le |\langle x, y \rangle| |\langle y, z \rangle| \le |\langle x, y \rangle| \langle x, z \rangle \le \langle x, z \rangle$$

where the last inequality follows since $|\langle x, y \rangle| \le 1$ for unit vectors.

The equality conditions follow from examining when each inequality becomes equality in the chain above. □

We continue with the proof of Theorem 4.1.

*Proof.* The system of particles is governed by equations

$$\dot{x}_k = \frac{1}{Z_k} \sum_{j=1}^{k} e^{\beta \langle Qx_k, Kx_j \rangle} (x_j - \langle x_j, x_k \rangle x_k),$$

where we omit $t$ from the notation for simplicity.

This system is autonomous, so we first explore its critical points and their stability. For autonomous systems with established convergence, it is well-known that for any absolutely continuous initialization, the limiting point is strongly unstable with probability zero (see (Shub, 2013, Thm. III.7, Ex. III.3) and (Geshkovski et al., 2023b, Lemma B.1)). Note that the proof in Geshkovski et al. (2023b) is stated for gradient ascent dynamics but it readily extends to any smooth autonomous dynamics on a compact Riemannian manifold.

Define:

$$f_k(x) := \frac{1}{\sum_{j=1}^{k} e^{\beta \langle Qx_k, Kx_j \rangle}} \cdot \sum_{j=1}^{k} e^{\beta \langle Qx_k, Kx_j \rangle} (x_j - \langle x_j, x_k \rangle x_k)$$

We aim to (i) find stationary points $x$ where all $f_k(x) = 0$ and (ii) analyze eigenvalues of the Jacobian $\left(\frac{\partial f_k}{\partial x_j}\right)$ at said stationary points.

Any critical point must satisfy one of the following:

- $x_1 = \ldots = x_n = \xi$ for some $\xi \in S^{d-1}$

- There exists $s \in \{2, \ldots, n\}$ such that $x_1 = \ldots = x_{s-1} = \xi$ and $x_s = -\xi$

Indeed, if the first condition fails, consider the first token $x_s$ where $x_s \neq x_1$. Then $f_s(x) = 0$ implies $x_s - \langle x_s, \xi \rangle \xi = 0$, forcing $x_s = \pm \xi$ so that $x_s = -\xi$ since we required $x_s \neq x_1$.

Our goal is to show that stationary points of the second kind are limiting points with probability zero with respect to the initialization distribution. Observe that since the system formed by the first $s$ particles is independent of subsequent ones, it suffices to show:

$$\mathbb{P}((x_1, \ldots, x_{s-1}, x_s) \to (\xi, \ldots, \xi, -\xi)) = 0$$

Since $x_1(t) = x_1(0)$, this reduces to:

$$\mathbb{P}(x_1 = \xi, (x_2, \ldots, x_{s-1}, x_s) \to (\xi, \ldots, \xi, -\xi)) = 0$$

By the law of total probability, it suffices to show that for almost all $\xi \in S^{d-1}$:

$$\mathbb{P}_{x_2, \ldots, x_s | x_1 = \xi}((x_2, \ldots, x_{s-1}, x_s) \to (\xi, \ldots, \xi, -\xi)) = 0$$

We are left to the function $f_s$ around $(x_1, \ldots, x_{s-1}, x_s) = (\xi, \ldots, \xi, -\xi)$. Observe that

$$f_s(\xi, \ldots, \xi, x_s) = w(\xi, x_s)(\xi - \langle \xi, x_s \rangle x_s)$$

where

$$w(\xi, x_s) = \frac{(s-1)e^{\beta \langle Qx_s, K\xi \rangle}}{(s-1)e^{\beta \langle Qx_s, K\xi \rangle} + e^{\beta \langle Qx_s, Kx_s \rangle}} > 0.$$

Observe that the Jacobian $\left(\frac{\partial f_k}{\partial x_j}\right)$ is block lower triangular, with blocks given by $\frac{\partial f_k}{\partial x_k}$. We show below that $\frac{\partial f_s}{\partial x_s}$ has an eigenvalue with positive real part, which is sufficient to establish strong instability.

At $x_2 = \ldots = x_{s-1} = \xi$ and $x_s = -\xi$:

$$f_s(\xi, \ldots, \xi, x_s) = w(\xi, x_s)(\xi - \langle \xi, x_s \rangle x_s)$$

where

$$w(\xi, x_s) = \frac{(s-1)e^{\beta \langle Qx_s, K\xi \rangle}}{(s-1)e^{\beta \langle Qx_s, K\xi \rangle} + e^{\beta \langle Qx_s, Kx_s \rangle}} > 0.$$

The classical Jacobian in $\mathbb{R}^d$ is:

$$\frac{\partial}{\partial x_s} f_s(\xi, \ldots, \xi, x_s) = \left(\frac{\partial}{\partial x_s} w(\xi, x_s)\right)(\xi - \langle \xi, x_s \rangle x_s)^\top + w(\xi, x_s)\frac{\partial}{\partial x_s}(\xi - \langle \xi, x_s \rangle x_s)$$

Hence

$$\frac{\partial}{\partial x_s} f_s(\xi, \ldots, \xi, x_s)\big|_{x_s = -\xi} = w(\xi, x_s)\frac{\partial}{\partial x_s}(\xi - \langle \xi, x_s \rangle x_s)$$

The spherical Jacobian is obtained by projecting onto $\xi^\perp$ and given by

$$(\mathbf{I} - \xi\xi^\top)\frac{\partial}{\partial x_s} f_s(\xi, \ldots, \xi, x_s)\big|_{x_s = -\xi} = w(\xi, -\xi) \cdot (\mathbf{I} - \xi\xi^\top)$$

This linear operator acts on $\xi^\perp$ and has eigenvalues $w(\xi, -\xi)$ with multiplicity $d-1$, which are all real positive, confirming strong instability.

By the center-stable manifold theorem (Shub, 2013, Thm. III.7, Ex. III.3), if a point has at least one eigenvalue with positive real part, then: (i) The center-stable manifold $W_{loc}^{cs}$ has positive co-dimension and (ii) Points converging to this equilibrium must enter $W_{loc}^{cs}$ at some finite time and (iii) the set of such points has measure zero.

More precisely, if a trajectory $(x_2, \ldots, x_s)$ converges to $(\xi, \ldots, \xi, -\xi)$, then:

$$\exists m \in \mathbb{Z}_{\geq 0} : (x_2(m), \ldots, x_s(m)) \in W_{loc}^{cs}.$$

Since our flow is a diffeomorphism, the pre-image of $W_{loc}^{cs}$ is also a manifold of positive codimension. Therefore, the set of initial conditions leading to convergence to $(\xi, \ldots, \xi, -\xi)$ is contained in a countable union of measure-zero sets, making it a measure-zero set itself.

We continue with an induction on the number of particles to show that with probability one, $x_2, \ldots, x_n$ all converge to $\xi$.

For the base case $k = 2$, observe that $x_2$ converges to $\xi$ except when initialized unstable equilibrium $x_2(0) = -\xi$.

Assume next that $x_2, \ldots, x_{k-1} \to \xi$ with probability one so that for any $\varepsilon > 0$, there exists a time $T_0$ after which

$$\min_{j<k} \langle x_j, \xi \rangle > 1 - \varepsilon \quad \text{a.s.}$$

Consider:

$$\frac{d\langle x_k(t), \xi \rangle}{dt} = \left\langle \mathbf{P}_{x_k(t)} \left( \frac{1}{Z_k(t)} \sum_{j=1}^{k-1} e^{\beta \langle Q x_k(t), K x_j(t) \rangle} x_j(t) \right), \xi \right\rangle$$

$$= \frac{1}{Z_k(t)} \left( \sum_{j=1}^{k-1} e^{\beta \langle Q x_k(t), K x_j(t) \rangle} \left( \langle x_j(t), \xi \rangle - \langle x_j(t), x_k(t) \rangle \langle x_k(t), \xi \rangle \right) \right).$$

From Lemma 6 we get that $\langle x_j(t), \xi \rangle - \langle x_j(t), x_k(t) \rangle \langle x_k(t), \xi \rangle > 0$ if $\langle x_j, \xi \rangle > 0$ and $|\langle x_k, \xi \rangle| < \langle x_j, \xi \rangle$. In particular, we get that the time derivative above is positive after time $T_0$ whenever $-1 + \varepsilon < \langle x_k, \xi \rangle < 1 - \varepsilon$ since we are guaranteed that $\forall j < k, \langle x_j, \xi \rangle > 1 - \varepsilon > 0$ after that time. But from the center-stable theorem argument, we know that $\langle x_k, \xi \rangle$ does not converge to $-1$ so there exists a time $T_1 > T_0$ at which $\langle x_k, \xi \rangle > -1 + \varepsilon$. After this time, either $x_k$ gets closer to $\xi$ (positive derivative) until $\langle x_k, \xi \rangle = 1 - \varepsilon$ after which time, $\langle x_k, \xi \rangle \geq 1 - \varepsilon$ forever. Since this argument is valid, for all $\varepsilon > 0$, we get that $x_k \to \xi$.

By induction, all points converge to $\xi$ with probability one.

$\square$

## B.3  Spectra of V

Here we include a figure showing the spectra of value matrices of different heads of a pre-trained transformer. Notice that most of them have real eigenvalue corresponding to $\lambda_{\max}$, justifying our focus on the real case. Interestingly, there are heads with negative $\lambda_{\max}$, even though starting spectrum of a Gaussian matrix is far from the left half-plane. This could indicate that some properties of $\lambda_{\max} < 0$ are desirable in practice, suggesting to consider other ways to initialise of $V$ for training.

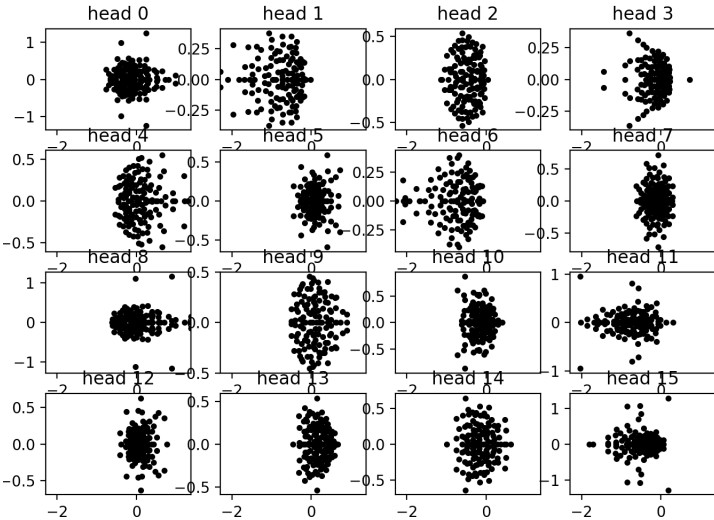

Figure 4: Eigenvalues of different heads of pre-trained albert-xlarge-v2 model in the complex plane.

# C  Meta-stable clustering

## C.1  Proof of Lemma 2

We prove the lemma for $d = 2$.

*Proof.* For $d = 2$, written in polar coordinates $x_k = \exp(i\varphi_k)$, the system is

$$\dot{\varphi}_k = \frac{1}{Z_k} \sum_{j=1}^{k-1} e^{\beta(\cos(\varphi_k - \varphi_j) - 1)} \sin(\varphi_j - \varphi_k), \tag{6}$$

with

$$Z_k = \sum_{j=1}^{k} e^{\beta(\cos(\varphi_k - \varphi_j) - 1)}.$$

Because of the $j = k$ term one has $Z_k \geq 1$, which implies

$$|\dot{\varphi}_{s_k}| \leq \sum_{j=1}^{s_k - 1} e^{\beta(\cos(\varphi_{s_k} - \varphi_j) - 1)} |\sin(\varphi_j - \varphi_{s_k})| = \sum_{j=1}^{s_k - 1} h(|\varphi_j - \varphi_{s_k}|).$$

Consider $\varphi_{r^*}$ to be the closest particle to $\varphi_{s_k}$ among previous ones, here $r^*$ depends on $s_k$ and $t$. Wlog, let us assume that $\varphi_{r^*} \in [\varphi_{s_k}, \varphi_{s_k} + \pi]$ and denote $\Delta = \varphi_{r^*} - \varphi_{s_k}$. We are going to show that $\Delta$ cannot decrease fast. At any time one has

$$-\dot{\Delta} = \dot{\varphi}_{s_k} - \dot{\varphi}_{r^*} = \frac{1}{Z_{s_k}} \sum_{j=1}^{s_k - 1} h(\varphi_j - \varphi_{s_k}) - \frac{1}{Z_{r^*}} \sum_{j=1}^{r^* - 1} h(\varphi_j - \varphi_{r^*}).$$

Let us bound both terms separately. Due to the definition of $\Delta$, and as long as $\Delta > \tau_{\beta}^*$ (from Lemma 4) we can bound the first term

$$|\dot{\varphi}_{s_k}| \leq s_k h(\Delta). \tag{7}$$

For the second one, we only leave the terms with $\varphi_j \in [\varphi_{r^*} - \pi, \varphi_{r^*}]$, because the other ones are negative

$$-\dot{\varphi}_{r^*} \leq \sum_{j=1}^{r^* - 1} |h(\varphi_j - \varphi_{r^*})| \mathbf{1}_{\varphi_j \in [\varphi_{r^*} - \pi, \varphi_{r^*}]}.$$

In other words, the only particles that drive $\varphi_{r^*}$ towards $\varphi_{s_k}$ are the ones that are in the same direction. However, there are no particles closer to $\varphi_{s_k}$ than $\varphi_{r^*}$. Therefore, being in the same direction implies being at least $2\Delta$-far away from $\varphi_{r^*}$.

If $h(2\Delta) < 0$, there are no particles $\varphi_j \in [\varphi_{r^*} - \pi, \varphi_{r^*}]$ and the sum is zero, otherwise we use Lemma 4 to upper bound each term with $h(2\Delta)$. Thus,

$$-\dot{\varphi}_{r^*} \leq s_k \max(h(2\Delta), 0) \leq s_k h(\Delta).$$

The last inequality follows from the fact that $h(\Delta) > 0$ and monotonicity of $h$ when $\Delta > \tau_{\beta^*}$. Combining the estimates, we obtain that for $\Delta > \tau_{\beta}^*$ one has

$$-\dot{\Delta} \leq 2 s_k h(\Delta).$$

To prove that for $t \in [0, T_k]$ we have $\Delta(t) > c\beta^{-1/2}$, we just need to verify that $c\beta^{-1/2} > \tau_{\beta^*}$ and that initially

$$\Delta(0) > c\beta^{-1/2} + 2 s_k T_k h(c\beta^{-1/2}),$$

which is true by definition of Rényi centers, and bounds on $c$ (1) and $T_k$ (2). Then, from (7) one has

$$\max_{t \in [0, T_k]} |\varphi_{s_k}(t) - \varphi_{s_k}(0)| < s_k T_k h(c\beta^{-1/2}) < \varepsilon c\beta^{-1/2}.$$

$\square$

## C.2 Proof of Lemma 3

*Proof.* We prove that the particles converge exponentially fast to some strongly stable critical point $\varphi^*$ via induction on their number.

*Induction base.* In this proof the induction base for $n = 1$ follows from the induction step proof applied to the first particle $\varphi_1$.

*Induction step.* Consider $\varphi_1(t), \ldots, \varphi_{n-1}(t)$. They do not depend on $\varphi_n$ and from induction hypothesis converge exponentially fast to some asymptotically stable point $\varphi_1^*, \ldots, \varphi_{n-1}^*$. In particular, one has $\dot{\varphi}_k \in L_1([0, \infty)), k \in [n-1]$.

Consider full derivative

$$\frac{dE_n(\varphi(t))}{dt} = \sum_{j=1}^{n-1} \frac{\partial E_n}{\partial \varphi_j} \dot{\varphi}_j(t) + \frac{\partial E_n}{\partial \varphi_n} \dot{\varphi}_n(t) = \sum_{j=1}^{n-1} \frac{\partial E_n}{\partial \varphi_j} \dot{\varphi}_j(t) - Z_n(t)|\dot{\varphi}_n(t)|^2.$$

Since all partial derivatives $\partial E_n / \partial \varphi_j$ are bounded on the compact manifold and all derivatives $\dot{\varphi}_j, j < n$ are in $L_1([0, \infty))$, one has

$$g(t) := \sum_{j=1}^{n-1} \frac{\partial E_n}{\partial \varphi_j} \dot{\varphi}_j(t) \in L_1([0, \infty)).$$

For the base case, i.e. $n = 1$, one simply has $g(t) \equiv 0$. Let us show that $f(t) := \frac{dE_n(\varphi(t))}{dt}$ is also in $L_1([0, \infty))$. Notice how $f(t)$ has a finite integral on any interval

$$\int_a^b f(t)dt = E_n(\varphi(a)) - E_n(\varphi(b)) < C < \infty \tag{8}$$

and is upper bounded

$$f(t) \leq g(t) \in L_1([0, \infty)).$$

Consider $f_+ = f(t)\mathbf{1}_{f(t)>0}$, $f = f_+ - f_-$. Clearly $f_+ \in L_1([0, \infty))$ as it is upper bounded by $|g| \in L_1([0, \infty))$. Moreover, the bound (8) shows that for all $T > 0$ one has

$$\int_0^T f_-(s)ds \leq \int_0^T f_+(s)ds + C < \|f_+\|_{L_1([0,\infty))} + C.$$

From this we get that $f_- \in L_1([0, \infty))$. Consequently, $f \in L_1([0, \infty))$. Since

$$Z_n(t)|\dot{\varphi}_n|^2 = g(t) - f(t) \in L_1([0, \infty)),$$

and $Z_n(t)$ has a uniform lower bound, we obtain $\dot{\varphi}_n \in L_2([0, \infty))$.

Let us show that $\dot{\varphi}_n$ is absolutely continuous. The trajectory of all particles $\varphi(t)$ is absolutely continuous, because $\dot{\varphi}(t)$ is bounded as a continuous vector field on a compact manifold. Then, $\dot{\varphi}_n(t)$ is absolutely continuous, because it is a composition of absolutely continuous vector field and an absolute continuous trajectory $\varphi(t)$.

Because $\dot{\varphi} \in L_2([0, \infty))$ and $\dot{\varphi}_n$ is absolutely continuous, it satisfies

$$\limsup_{t \to \infty} |\dot{\varphi}_n(t)| = 0.$$

In other words, because of the upper bound $Z_n(t) < C$, one has

$$\lim_{t \to \infty} \left| \frac{\partial}{\partial \varphi_n} E_n(\varphi_1(t), \ldots, \varphi_n(t)) \right| = 0. \tag{9}$$

Finally, we are going to prove that $\varphi_n$ converges to some critical point $\varphi_n^*$ with

$$\frac{\partial}{\partial \varphi_n} E_n(\varphi_1^*, \ldots, \varphi_n^*) = 0.$$

Consider the set $\mathcal{E} = \{\psi : \frac{\partial}{\partial \varphi_n} E(\varphi_1^*, \ldots, \varphi_{n-1}^*, \psi) = 0\}$. By assumption, the energy function $E_n(\varphi_1^*, \ldots, \varphi_{n-1}^*, \psi)$ has isolated critical points w.r.t. $\psi$, thus the set of zeroes $\mathcal{E}$ is a finite collection of points $\mathcal{E} = \{\psi_1, \ldots, \psi_m\}$.

When $\varphi_1, \ldots, \varphi_{n-1}$ converge to $\varphi_1^*, \ldots, \varphi_{n-1}^*$ the set

$$\left\{ \varphi_n : \left| \frac{\partial}{\partial \varphi_n} E(\varphi_1, \ldots, \varphi_n) \right| < \varepsilon \right\}$$

is inside a collection of distinct intervals around $\psi_1, \ldots, \psi_m$ for small enough $\varepsilon$. Therefore, due to (9), from some moment $\varphi_n(t)$ stays only in one of those intervals. They collapse into points as we take $\varepsilon \to 0$ and $t \to \infty$, proving that $\varphi_n$ converges to some $\varphi_n^* \in \mathcal{E}$. It remains to prove that observed convergence is to a strongly stable point.

We know that the limiting point is critical and for almost any initialisation it is not strongly unstable. This is a well-known fact that for autonomous systems convergence to a strongly unstable point happens with probability zero, that we used in Section B.2. One could find it in (Geshkovski et al., 2023b, Lemma B.1), based on the center manifold theorem (Shub, 2013, Thm. III.7, Ex. III.3). Therefore, from the assumption on critical points, the limiting point is strongly stable. Then, the convergence happens exponentially fast, because locally the whole neighbourhood of a strongly stable point is its stable manifold $W_{loc}^s$, see (Shub, 2013, Thm. III.7, Ex. III.3). □

### C.3 Proof of Theorem 5.1

*Proof.* Our prove consists of two parts. In order to obtain convergence we are going to apply Lemma 3. The system has exactly gradient-like form for energy functionals

$$E_k(\varphi_1, \ldots, \varphi_k) = -\left( \sum_{j=1}^{k-1} e^{\beta(\cos(\varphi_k - \varphi_j) - 1)} + \sum_{j=1}^{m} a_j e^{\beta(\cos(\varphi_k - \theta_j) - 1)} \right)$$

and bounded normalization factors

$$Z_k(\varphi_1, \ldots, \varphi_k) = \sum_{j=1}^{k} e^{\beta(\cos(\varphi_k - \varphi_j) - 1)} + \sum_{j=1}^{m} a_j e^{\beta(\cos(\varphi_k - \theta_j) - 1)}.$$

Therefore, if conditions of Lemma 3 are satisfied, we have convergence to a strongly stable critical point. In that case, it is sufficient to prove that for any strongly stable critical point $\varphi^*$, all particles $\varphi_k^*$ are $\varepsilon \beta^{-1/2}$-close to one of the centers $\theta_j$. We show this via induction on $k$, because new particles do not affect the movement of the previous ones.

To justify our use of Lemma 3, we need to check that all critical points are either strongly stable or strongly unstable, i.e. that the Jacobian at any critical point with only non-positive eigenvalues actually has no zero eigenvalues. Moreover, for our conclusion, we need to check that if all the eigenvalues are negative, then the critical point is clustered around $\theta$. Let us start with that.

In what follows we repeatedly use properties of $h$ and $g$ from Lemma 4.

The system is of the form

$$\dot{\varphi}_k = -\frac{1}{Z_k} \left( \sum_{j=1}^{k-1} h(\varphi_k - \varphi_j) + \sum_{j=1}^{m} a_j h(\varphi_k - \theta_j) \right) =: f_k(\varphi, \theta).$$

Since $f_k$ does not depend on $\varphi_{k+1}, \ldots, \varphi_n$, the Jacobian $(\frac{\partial f_k}{\partial \varphi_j})$ is lower-triangular. Therefore, its eigenvalues are $\frac{\partial f_k}{\partial \varphi_k}$. Let us assume that all of the eigenvalues are non-positive. Since at a critical point $f_k(\varphi^*, \theta)$ itself is zero, one has

$$\frac{\partial f_k}{\partial \varphi_k}(\varphi^*, \theta) = -\frac{1}{Z_k} \left( \sum_{j=1}^{k-1} g(\varphi_k^* - \varphi_j^*) + \sum_{j=1}^{m} a_j g(\varphi_k^* - \theta_j) \right) \leq 0.$$

This implies that one of the terms $g(\varphi_k^* - \varphi_j^*)$ or $g(\varphi_k^* - \theta_j)$ is positive. From properties of $g$ in Section A.2, we obtain that $\varphi_k^*$ is $\tau_\beta^*$-close to either one of the centers $\theta_j$ or one of the previous particles $\varphi_j^*$. By induction hypothesis, all previous particles are $\varepsilon \beta^{-1/2}$-close to the centers, i.e. $\varphi_k^*$ is in $\tau_\beta^* + \varepsilon \beta^{-1/2}$-neighbourhood of some center, wlog $\theta_1$.

Let us assume that $\varphi_k^*$ is $\varepsilon\beta^{-1/2}$-far from $\theta_1$. From criticality of the point, it is true that

$$\sum_{j=1}^{k-1} h(\varphi_k^* - \varphi_j^*) + \sum_{j=1}^{m} a_j h(\varphi_k^* - \theta_j) = 0,$$

By induction hypothesis, all previous particles are $\varepsilon\beta^{-1/2}$-close to some center. Denote the set of particles that are close to $\theta_1$ as $S$. Then the criticality can be written as

$$a_1 h(\varphi_k^* - \theta_1) + \sum_{j \in S} h(\varphi_k^* - \varphi_j^*) = -\sum_{j \notin S} h(\varphi_k^* - \varphi_j^*) - \sum_{j=2}^{m} a_j h(\varphi_k^* - \theta_j).$$

All of the terms on the left hand side have the same sign, because $\varphi_j^*, j \in S$ are $\varepsilon\beta^{-1/2}$-close to $\theta_1$ and $\varphi_k^*$ is $(\tau_\beta^* + \varepsilon\beta^{-1/2})$-close to $\theta_1$. Any other particle/center is at least $(c - 2\varepsilon)\beta^{-1/2} - \tau_\beta^*$ far from $\varphi_k^*$, because centers are $c\beta^{-1/2}$-separated. The absolute value of the l.h.s. can be lower bounded

$$|a_1 h(\varphi_k^*) + \sum_{j \in S} h(\varphi_k^* - \varphi_j^*)| \geq h(\varepsilon\beta^{-1/2}).$$

The absolute value of the r.h.s. can be upper bounded

$$|\sum_{j \notin S} h(\varphi_k^* - \varphi_j^*) + \sum_{j=2}^{m} a_j h(\varphi_k^* - \theta_j)| \leq N h((c - 2\varepsilon)\beta^{-1/2} - \tau_\beta^*) < N h((c - 1 - 2\varepsilon)\beta^{-1/2}),$$

where in the last part we used the fact from Section A.2 that $\tau_\beta^* < \beta^{-1/2}$.

Therefore, one has

$$N h((c - 1 - 2\varepsilon)\beta^{-1/2}) \geq h(\varepsilon\beta^{-1/2}),$$

which contradicts our assumption on system parameters.

It remains to check that when the eigenvalues of the Jacobian are not positive, they are actually strictly negative. We already know that in that case all the particles are clustered around centers $\theta_j$. Let us assume that

$$\frac{\partial f_k}{\partial \varphi_k}(\varphi^*, \theta) = 0.$$

We know that $\varphi_k^*$ is $\varepsilon\beta^{-1/2}$-close to some center, wlog $\theta_1$. Therefore,

$$g(\varphi_k^* - \theta_1) \geq g(\varepsilon\beta^{-1/2}).$$

On the other hand, the only negative terms in the sum

$$\sum_{j=1}^{k-1} g(\varphi_k^* - \varphi_j^*) + \sum_{j=1}^{m} a_j g(\varphi_k^* - \theta_j) = 0$$

are from particles that are not in the neighbourhood of $\theta_1$, so they are at least $(c - 2\varepsilon)\beta^{-1/2}$-far from $\varphi_k$. Therefore, for the negative terms to balance the positive $g(\varphi_k^* - \theta_1)$, it should be true that

$$g(\varepsilon\beta^{-1/2}) < -N g((c - 2\varepsilon)\beta^{-1/2}).$$

This contradicts the parameters choice.

$\square$

## C.4 Rényi centers vs strong Rényi centers

In this section we estimate the number of clusters our approach predicts. As a reminder, for a sequence of particles $X_i$ on a unit sphere, and geodesic distance dist, we consider two subsequences

- Rényi centers is $X_{s_j}, j \geq 1$ such that $\forall j \min_{i < j} \text{dist}(X_{s_j}, X_{s_i}) > \delta$,
- strong Rényi centers is $X_{s_j}, j \geq 1$ such that $\forall j \min_{i < s_j} \text{dist}(X_{s_j}, X_i) > \delta$.

Estimating the number of elements in the first sub-sequence is well-known as famous Rényi parking problem Renyi (1958). In the case $d = 2$, the result from Dvoretzky and Robbins (1964) implies that as $\delta \to 0$, the average number of elements in the sequence approaches $c2\pi/\delta$ superexponentially fast, where $c \approx 0.75$ is the Rényi constant. However, this problem becomes significantly harder in higher dimensions. In contrast, it is easy to compute the average number of elements in the second sequence in any dimension and even for a wider class of distributions.

**Lemma 7.** *In an infinitely long sequence $X_i$ the average number of variables $X_{s_j}$ chosen by strong Rényi parking is the inverse spherical cap surface area $1/\sigma^{d-1}(B_\delta)$. In particular, it grows as $1/\delta^{d-1}$ with dimension and can be computed directly in lower dimensions*

$$1/\sigma^{d-1}(B_\delta) = \begin{cases} \pi\delta^{-1}, d = 2 \\ (3\sin^2(\delta/2))^{-1}, d = 3 \end{cases}$$

*Proof.* Consider a sequence of *i.i.d.* points $X_i$ being sampled on a $d$-dimensional sphere $\mathbb{S}^{d-1}$ according to some distribution $\mu$. Let us find the probability that $k$-th particle $X_k$ is chosen by strong Rényi parking with distance $\delta$. This event can be written as

$$\mathbb{P}[\cap_{j=1}^{k-1}\{\text{dist}(X_k, X_j) > \delta\}] = \int_{\mathbb{S}^{d-1}} \mathbb{P}[\cap_{j=1}^{k-1}\{\text{dist}(x, X_j) > \delta\}]d\mu(x),$$

where we used total probability formula. Then, since $X_j$ are i.i.d., we can write it as

$$\int_{\mathbb{S}^{d-1}} \mathbb{P}(\text{dist}(x, X_1) > \delta)^{k-1}d\mu(x) = \int_{\mathbb{S}^{d-1}} (1 - \mu(B_\delta(x)))^{k-1}d\mu(x).$$

From this we obtain that the average number of chosen points is equal to

$$\int_{\mathbb{S}^{d-1}} \sum_{k=1}^{\infty} \mathbb{P}(\text{dist}(x, X_1) > \delta)^{k-1}d\mu(x) = \int_{\mathbb{S}^{d-1}} \frac{1}{\mu(B_\delta(x))}d\mu(x) = \frac{1}{\sigma^{d-1}(B_\delta)},$$

where the last equality is correct for any spherically harmonic distribution $\mu$, in particular for a uniform measure. $\square$

