# OpenReview forum: "Clustering in Causal Attention Masking"
_NeurIPS.cc/2024/Conference — NeurIPS 2024 poster_

### Official Review · Reviewer_Apon · 2024-07-09

**Soundness:** 3
**Presentation:** 3
**Contribution:** 3
**Rating:** 7
**Confidence:** 3

**Summary:**

This paper studies the representations or tokens generated by a sequence of causal attention layers. To this end, and following the example of prior works, the authors model such a sequence as a discretization of a system of ODEs. Each token in the input sequence is modeled as a particle and the evolution of each token with depth is modeled as an interacting particle system. Layer normalization is used to ensure each token lies on he unit sphere. A number of theoretical results are derived and conjectures made in this setting (referred to as CSA dynamics).
- Lemma 1 roughly characterizes rate at which tokens approache the leading eigenspace of the value matrix.

- Theorem 4.1 characterizes a simple setting in which tokens collapse to a single point asymptotically.

- A number of conjectures are made based on the dynamics as well as experiments describing situations in which tokens or particles spread out or collapse onto a few discrete points.

- Meta stable clusters (which persist for a significant time but disappear eventually) are also studied in a simplified 2 dimensional setting and an interesting connection to the Renyi parking problem is made.

Perhaps the key takeaway is that each token (or particle) is driven by an internal force as well as an external force, which is either attractive or repulsive according to the sign of the largest eigenvalue of the value matrix. This in turn controls the diversity of token representations asymptotically.

**Strengths:**

- Originality: not aware of other similar analyses for causal attention, the modified dynamics due to the causal masking also make the analysis challenging.

- Quality and clarity: paper is well written, motivated and clear.

- Significance: there are some interesting takeaways, e.g., the role of the largest eigenvalue in driving particles to be diverse versus collapsing onto a few points as well as the implications for dimensionality reduction.

**Weaknesses:**

- Many results are either asymptotic in nature, this could limit their relevance for explaining phenomena observed in practice.

- The study of meta clustering is restricted to the 2D setting.

- One might argue that there are a number of other aspects which hinder drawing practical takeaways, for instance weights are tied across different layers, no MLP layers between attention layers etc.

- I don't see anywhere a discussion of the differences in the token dynamics + asymptotics of causal attention versus standard self attention, which would seem a natural and useful thing to include.

**Questions:**

The main thing I think it would be nice to see more discussion of is how the restriction to causal attention impacts the dynamics: can you highlight any important differences between the dynamics of the tokens of causal versus standard self-attention?

**Limitations:**

Yes

---

> ### Author Rebuttal · Authors · 2024-08-07
>
> We are grateful to the reviewers for their positive feedback. We appreciate that they acknowledged the novelty and significance of our setting and are glad that they enjoyed our exposition **Quality and clarity: the paper is well-written, motivated, and clear**.
>
> **Many results are either asymptotic in nature, this could limit their relevance for explaining phenomena observed in practice.**
> This is true for final configurations and most of the results, but the meta-clustering section is devoted to the study of a phenomenon that is significant for practice. As mentioned in Section 6, it lacks a complete non-asymptotic proof, but it is a step forward toward better understanding what is happening in intermediate times that are practically relevant (see Fig. 2).
>
> **The study of meta-clustering is restricted to the 2D setting.** Although the restriction to the 2D setting is made, this is an important foundational step for two reasons. First, the interaction force is the same in any dimension, and so we believe that it is possible to lift the $d = 2$ assumption in the future. Secondly, because of the dimension reduction that matrix $V$ introduces, the lower-dimensional case $d = 2$ is actually relevant for the general case too.
>
> **The main thing I think it would be nice to see more discussion of is how the restriction to causal attention impacts the dynamics: can you highlight any important differences between the dynamics of the tokens of causal versus standard self-attention?**
> Thank you for this question. We are going to include an overview of key differences and compare our results with what was already known for the non-causal case. To summarize the key differences: in Geshkovski et al. (2023), the authors analyze non-causal attention by utilizing its gradient descent structure when $QK^T = V$. The part of their work that is connected to ours is the clustering part, where they prove asymptotic convergence to a single cluster under additional dimension/temperature restrictions. These restrictions were significantly lifted in [Criscitiello et al. (2024)](https://arxiv.org/abs/2405.18273) to $d \geq 3$ and $\beta \leq 1$, but they are still concerned with non-causal dynamics (and utilize gradient descent structure). In our work, we study causal attention in the same framework, but there is no gradient descent structure, which leads to different analysis techniques and different restrictions. In particular, that is why we are able to show asymptotic convergence to a single cluster for arbitrary matrices $Q$, $K$, and $V = I_d$, and conjecture final configurations for different choices of $V$ (ideologically, those conjectures follow from our proof of Theorem 4.1, but there are significant technicalities that are yet to be resolved). From our understanding, the final configurations in causal and non-causal cases are similar. The key difference is in the more practically relevant meta-clustering phenomenon. To the best of our knowledge, there is no successful theoretical approach to meta-clustering; in [Geshkovski et al.] it is noticed for original non-causal dynamics and it is conjectured that the number of meta-clusters should be of order $\sqrt{\beta}$ in the 2-dimensional case. In our work, we introduced a unique perspective on the appearance of meta-clusters in causal attention through Rényi parking, which predicts the conjectured $\sqrt{\beta}$ clusters, and showed relevant clustering results (mainly Theorem 5.1). Here, the difference is that in the causal case we are able to predict where meta-clustering is going to occur, while for standard non-causal dynamics it seems to be more complicated and it is yet to be done.

---

> > ### Comment · Reviewer_Apon · 2024-08-08
> >
> > Thanks for your response, I'll keep my score.

---

### Official Review · Reviewer_9pe5 · 2024-07-10

**Soundness:** 2
**Presentation:** 2
**Contribution:** 2
**Rating:** 5
**Confidence:** 2

**Summary:**

This paper strengthens the theoretical results from prior work by presenting causally masked attention used in AIGC.

The authors prove asymptotic convergence to a single cluster for arbitrary key-query matrices and an identity value matrix under causal self-attention. This significantly extends the results of previous studies.

**Strengths:**

This paper provide novel insight in understanding causal attention mechanisms, proposing new mathematical models and proof techniques that extend existing knowledge.

By linking the study to the Rényi parking problem, the authors provide a unique perspective on clustering phenomena in self-attention mechanisms.

**Weaknesses:**

This paper is really hard to follow, Geshkovski et al. (2023c) is referenced numerous times throughout the article, even including in the abstract. The authors should clearly summarize the previous work, and then emphasize their own contributions building on that foundation.

While the paper extends the understanding of causal attention mechanisms, more empirical evidence is needed to validate the results across a wider range of scenarios and applications.

**Questions:**

This work largely builds upon the framework of Geshkovski et al. (2023c). Could the author give a more concise statement to emphasize the differences and contributions in this paper?

Could the author provide more practical examples or evidence to prove the superiority of this theory?

**Limitations:**

This paper has introduces the limitations.

---

> ### Author Rebuttal · Authors · 2024-08-07
>
> We thank the reviewer for their feedback. We are glad that they noted our perspective on the clustering phenomena in transformers **By linking the study to the Rényi parking problem, the authors provide a unique perspective on clustering phenomena in self-attention mechanisms** and the theoretical novelty of the paper **This paper provides novel insights in understanding causal attention mechanisms, proposing new mathematical models and proof techniques that extend existing knowledge.**
>
> **This paper is really hard to follow, Geshkovski et al. (2023) is referenced numerous times throughout the article, even including in the abstract. The authors should clearly summarize the previous work, and then emphasize their own contributions building on that foundation.** This is a great point that we are going to address by providing a supplementary paragraph on how the dynamical system is obtained from the causal transformer architecture for a self-contained introduction to the topic.
>
> **This work largely builds upon the framework of Geshkovski et al. (2023). Could the authors give a more concise statement to emphasize the differences and contributions in this paper?**
> To summarize the key differences: in Geshkovski et al. (2023), the authors analyze non-causal attention by utilizing its gradient descent structure when $QK^T = V$. The part of their work that is connected to ours is the clustering part, where they prove asymptotic convergence to a single cluster under additional dimension/temperature restrictions. These restrictions were significantly lifted in [Criscitiello et al. (2024)](https://arxiv.org/abs/2405.18273) to $d \geq 3$ and $\beta \leq 1$, but they are still concerned with non-causal dynamics (and utilize gradient descent structure). In our work, we study causal attention in the same framework, but there is no gradient descent structure, which leads to different analysis techniques. In particular, that is why we are able to show asymptotic convergence to a single cluster for arbitrary matrices $Q$, $K$, and $V = I_d$, and conjecture final configurations for different choices of $V$ (ideologically, those conjectures follow from our proof of Theorem 4.1, but there are significant technicalities that are yet to be resolved). Moreover, to the best of our knowledge there is no theoretical approach to meta-clustering, in [Geshkovski et al.] it is only conjectured that the number of meta-clusters should be of order $\sqrt{\beta}$ in the 2-dimensional case. In our work, we introduced a unique perspective on the appearance of meta-clusters in causal attention through Rényi parking, which predicts the conjectured $\sqrt{\beta}$ clusters, and showed relevant clustering results (mainly Theorem 5.1).
> We are going to add a similar comparison in the paper.
>
> **While the paper extends the understanding of causal attention mechanisms, more empirical evidence is needed to validate the results across a wider range of scenarios and applications.**
> Extensive experiments with real-life models are very interesting and our theory suggests several insights to test. However, as the paper is already heavily theoretical, we believe that any proper experiments are out of scope of this work and a prospect for future research.

---

### Official Review · Reviewer_Bsfq · 2024-07-11

**Soundness:** 3
**Presentation:** 3
**Contribution:** 3
**Rating:** 6
**Confidence:** 2

**Summary:**

This work extends the work by Geshkovski et al. 23c, which analyzes the mean-field gradient flow of Transformer models and shows the emergence of clusters with full self-attention, to the ones with causal self-attention.
Transformer with causal self-attention is modeled as an interacting-particle system on the sphere, as tokens are normalized by Layer Normalization.
The authors conjecture that the largest eigenvalues of the value matrices alone governs the final states of the token particles.
Finally, the problem is connected with the Rényi parking process to show that the particles of causal self-attention Transformer reach metastable clusters under certain conditions.

**Strengths:**

- The authors extend the analysis of full self-attention Transformers as interacting-particle systems to causal self-attention Transformers.
As the current success of Transformers is mainly attributed to the autoregressive ones, the analysis of such models is relevant to the community.
- They connect the problem with the Rényi parking process and then show that, when the weight matrices are identity in 2D, the particles reach metastable clusters (Theorem 5.1).
- I appreciate that they discuss limitations using a single independent section.
- They presented many figures from numerical experiments, which helped me understand the work.

**Weaknesses:**

- Although the results are interesting, the most exciting parts are conjectures, and the proven results are under strict conditions (e.g., Theorem 4.1 and Theorem 5.1).

**Questions:**

- What are the practical implications of the results?

**Limitations:**

The authors discuss the limitations of the work (Section 6).

# Suggestions

- The references are not well maintained. Geshkovski et al. 2023a and Geshkovski et al. 2023b are identical, and it is accepted at NeurIPS 2023.
- $\textsf{dist}$ in L 109 seems to be defined in L 228.

---

> ### Author Rebuttal · Authors · 2024-08-07
>
> We thank the reviewer for their positive feedback. We are encouraged that they appreciate the relevance of our work's subject: **The authors extend the analysis of full self-attention Transformers as interacting-particle systems to causal self-attention Transformers. As the current success of Transformers is mainly attributed to the autoregressive ones, the analysis of such models is relevant to the community.**
>
> Let us address the weakness **Although the results are interesting, the most exciting parts are conjectures, and the proven results are under strict conditions (e.g., Theorem 4.1 and Theorem 5.1)** by discussing the conditions. Theorem 4.1 is limited to the case $V=I_d$. Firstly, it is already a generalization of the results by [Geshkovski et al. (2023)](https://arxiv.org/abs/2312.10794) and [Criscitiello et al. (2024)](https://arxiv.org/abs/2405.18273) for non-causal models. This is because they utilize a gradient descent approach to convergence, which is limited to the case $QK^T = V$. In this regard, our result shows that matrices $Q$ and $K$ do not affect final convergence. Secondly, the provided proof is more general and could be followed to argue that the conjectured results for $V$ are true. The problem of the appearance of unstable critical curves that arises there is mostly technical, not ideological, but it requires further thorough research. Theorem 5.1 is limited to $V = I_d$ for a reason. There is numerical evidence and an argument in Section 3, mentioned on line 218, that we expect that tokens quickly project to the subspace of a few top eigenvectors. This is why, from that point of view, $V = I_d$ in low dimension is a foundational case and an important part of any system. The biggest obstacle in meta-clustering is its non-asymptotic nature, which makes it hard to study. That is why we assume the simplest $Q=K=V=I_2$ setting for two reasons: this assumption allows us to focus on the hardest part (meta-clustering itself), and it is still an important step for general dynamics because of the heuristic dimension reduction that is yet to be proven.
>
> **What are the practical implications of the results?**
>
> Relevant practical implications are yet to be tested, but there are several promising avenues. Clustering has been observed numerically, but it is poorly understood what mechanisms are underlying its occurrence in transformers. There is an empirical study that suggests that clustering might correspond to transformers learning specific tasks/concepts and assigning them positions in the representation space. Then, being able to control the appearance of these meta-clusters through parameters and predict their appearance might help construct better models for specific tasks.

---

### Official Review · Reviewer_6isq · 2024-07-11

**Soundness:** 3
**Presentation:** 4
**Contribution:** 3
**Rating:** 7
**Confidence:** 1

**Summary:**

This paper presents a theoretical framework where causal attention masking can be recast into an interacting particle system. The authors start by introducing the dynamics of the first token and extend it to $n$ tokens. They then discuss the token configurations as $t \rightarrow \infty$ (i.e., infinite number of attention layers) when $V = I_d$ and make conjectures for more general cases where $V \neq I_d$. Lastly, the authors discuss the discovery of meta-clustering in Geshkovski et al. and adapt causal attention to this framework using the definition of Rényi centers.

**Strengths:**

•	Exploring the theoretical aspects behind the full attention and causal attention mechanisms is a very important topic in our understanding of how Transformers and modern LLM/LMMs work.

•	The authors give a comprehensive overview of the background and make a smooth transition to the token dynamics in causal attention.

•	The authors clearly explain the dynamics and the final states of the tokens with visualizations.

•	Despite the paper’s extensive theoretical and mathematical details, its main narrative is clear and easy to understand.

**Weaknesses:**

•	Although the authors argue about the complexity of the problem, $V = I_d$ might be too limited for real-world use cases.

•	The probability measure $\mu_0$ is first mentioned in Conjecture 1, but it is not clearly defined until Theorem 5.1. Same for the geodesic distance $dist$, which is first introduced in Lemma 1 but is not defined until Section 5.1.

•	Small typo: Line 140, To get a better grasp of the effects of “how” the external force works, … Line 186, in the full-attention dynamics, …

**Questions:**

It is well-known that transformer-based language models produce more stochastic outputs at higher temperatures $\beta$. What additional insights does meta-clustering provide beyond this established understanding? Does clustering of the tokens indicate similar outputs?

**Limitations:**

The authors have provided a relatively comprehensive discussion of limitations in Section 6.

---

> ### Author Rebuttal · Authors · 2024-08-07
>
> We thank the reviewer for the positive feedback. We are glad they think that **exploring the theoretical aspects behind the full attention and causal attention mechanisms is a very important topic in our understanding of how Transformers and modern LLMs/LMMs work** and that they enjoyed our presentation.
>
> To address the biggest mentioned weakness, **although the authors argue about the complexity of the problem, it might be too limited for real-world use cases**, we note that even though on paper $V = I_d$ is a significant limitation, it is not irrelevant for practice. This is largely because of the dimensionality reduction effect, which is justified by numerical experiments and can be seen as an effect of the matrix $V$, as discussed in Section 3 and mentioned in line 219. We expect that for the general case of $V$, all particles quickly converge to the subspace spanned by a few top eigenvectors, where the matrix action is close to the identity (see Fig. 1b for an example). Thus, understanding the behaviour of the system for $V = I_d$ (and even in low-dimensional cases as Theorem 5.1 does) is important for understanding the general picture as well.
>
> We will add earlier definitions of $\mu_0$ and $\textrm{dist}$ and correct the mentioned typos. Thank you for pointing them out.
>
> **It is well-known that transformer-based language models produce more stochastic outputs at higher temperatures. What additional insights does meta-clustering provide beyond this established understanding? Does clustering of the tokens indicate similar outputs?**
>
> In addition to the mentioned knowledge, our observation might lead to several additional insights. For example, the observed connection to the R\'enyi parking problem allows us to connect the number of meta-clusters with temperature and effective dimension. Moreover, there is an empirical study, that suggests that clustering corresponds to transformers learning specific tasks/concepts and assigning them positions in the representation space. Then, being able to control the appearance of these meta-clusters through parameters and predict their appearance might help construct better models for specific tasks. To our best understanding, in practical transformers tokens exhibit clustering, but do not converge to the same point completely (as one might expect from Theorem 4.1). This is an effect of different weights/MLP layers, but it also can be seen in our Theorem 5.1 -- with some tokens fixed, convergence happens not to the centers themselves, but somewhere close to them. Then, the outputs are not going to be completely the same, but arguably correspond to the same topic/task depending on the field of use. This is a really interesting question, how clustering corresponds to produced outputs, that requires further research.

---

> > ### Comment · Reviewer_6isq · 2024-08-10
> > **Response to rebuttal**
> >
> > Thank you for your comprehensive responses to my comments. I will keep my score for now.

---

### Decision · Program_Chairs · 2024-09-25

**Decision:**

Accept (poster)

**Comment:**

The paper studies clustering dynamics in causal transformers, following up on previous work that considered non-causal transformers. The reviewers found the theory interesting and insightful, though with unclear connections to practice. Overall, the consensus was positive as the proposed theory seems relevant for the community, and the paper is well written. We therefore recommend acceptance as a poster.